# Insomnia, Time Perspective, and Personality Traits: A Cross-Sectional Study in a Non-Clinical Population

**DOI:** 10.3390/ijerph191711018

**Published:** 2022-09-03

**Authors:** Marco Fabbri, Alessia Beracci, Monica Martoni

**Affiliations:** 1Department of Psychology, University of Campania “Luigi Vanvitelli”, 81100 Caserta, Italy; 2Department of Psychology, University of Granada, 18011 Granada, Spain; 3Department of Experimental, Diagnostic and Specialty Medicine (DIMES), Alma Mater Studiorum, University of Bologna, 40126 Bologna, Italy

**Keywords:** insomnia, time perspective, personality traits, conscientiousness, extraversion, emotional stability, future, past-negative, present-hedonistic, mediation analysis

## Abstract

Insomnia disorder is considered a public health problem and additional studies should investigate predisposing and perpetuating factors. This study examined the relationship between Big Five personality traits, time perspective, and insomnia. In a cross-sectional study, 400 participants (227 women; age range 18–74 years) were administered the Big Five Inventory—10 items, the Zimbardo Time Perspective Inventory, and the Insomnia Severity Index (ISI). A measure of chronotype was also included for control purposes. The results show that insomniacs reported lower scores for conscientiousness and extraversion, and for past-positive (PP) and future (F) perspectives, whereas they obtained higher scores for past-negative (PN) perspectives and deviation from a balanced time perspective. The correlations confirmed these findings, but negative correlations between present-hedonistic (PH) perspective and ISI score, and between emotional stability and ISI score, were also found. The mediation analyses showed that F played an indirect role in the relationship between consciousness and ISI score, PN had an indirect effect on the relationship between emotional stability and ISI or between extraversion and insomnia, and PH had an indirect effect on the relationship between extraversion and ISI score. The current outcomes shed light on the mechanisms which serve to mediate the relationship between insomnia and personality traits.

## 1. Introduction

Insomnia disorder is characterized by difficulty falling asleep, frequent nighttime awakenings with difficulty returning to sleep, and/or awakening earlier in the morning than desired [1,2]. This sleep disorder is also characterized by distress or impairment in functioning and daytime sleepiness, fatigue, impairment in cognitive performance, and mood disturbances. Because of the characteristics related to insomnia, this disorder is considered a major public health problem, especially considering the estimation of insomnia rates [3,4,5] in the general population, and it is both a physically and an economically demanding disorder. Linked to these points, patients with insomnia make greater use of healthcare services, with an increased frequency of medical visits [6,7]. Thus, research investigating which factors are associated (or predicted) with insomnia seems to be relevant for public health.

Personality traits have been theorized as a predisposing and potentially perpetuating factor for this sleep disorder [8]. For instance, several studies have shown that insomnia is related to a greater degree of maladaptive perfectionistic traits, neuroticism, negative affect, social inhibition, internalization, and anxious concerns, together with lower levels of conscientiousness and emotional stability [9,10,11,12,13]. When the Big Five model (BFM) [14] of personality is considered, it suggests a relationship between insomnia and a low level of conscientiousness on one hand and, on the other hand, between insomnia and a high degree of neuroticism (i.e., emotional stability factor of BFM) [12,15,16,17,18,19,20,21]. Indeed, conscientious individuals are responsible, dependable, motivated, have higher levels of self-control, regulate their impulses better and avoid all sleep-impairing behaviors (e.g., caffeine). At the same time, neurotic individuals tend to experience high levels of stress, have difficulties with emotion regulation with a high risk of increased negative affect and poor effortful control, and tend to cope with their poor nighttime sleep and daytime sleepiness using caffeine, alcohol, or other maladaptive behaviors, thus, decreasing sleep hygiene and perpetuating this destructive cycle [9,10,11,12,13,15,16,17,18,19,20,21]. However, the results supporting the relationship between insomnia and personality traits remain mixed and inconclusive, probably due to the role of other variables that mediate the association between personality and insomnia [9,11,12,13,21].

In the present study, we proposed the time perspective (TP) as an emerging variable in the relationship between personality traits and insomnia. TP indicates an individual’s habitual way of relating to their personal past, present, and future [22], and the Zimbardo Time Perspective Inventory (ZTPI) [22] is widely used to capture individual differences for five TPs (i.e., encompassing five subscales). To the best of our knowledge, few studies have investigated the relationship between TP and (poor) sleep quality. In one study, Vranesh et al. [23] reported positive associations between all past, present, and future TPs and the sleep quality in young students. Specifically, high past-negative (PN: a negative and aversive view of the past); high present-hedonistic (PH: an attitude toward the present involving immediate pleasure-seeking with little consideration of future consequences); high present-fatalistic (PF: a hopeless and helpless view of the present, where present behavior is considered as irrelevant to future consequences); or low future (F: a broad orientation toward the future involving optimism and striving for future goals and rewards) temporal orientations were associated with poor sleep quality. However, Vranesh et al. [23] also reported an inexplicable positive association between past-positive (PP: a positive, warm, and nostalgic view of the past) and poor sleep quality, requiring further investigations. Using a modified version of the ZTPI (with six subscales), Rönnlund and Carelli [24], in a study in older adults, reported that PN (and additional future-negative subscale or FN: an aversive view of the future with negative expectations and worry of future consequences) was predictive of poor sleep quality, with a small additional impact of PF, clarifying the relationship between TP and sleep quality. In a subsequent study, Rönnlund et al. [25] reported that a lower level of sleep quality was predicted by PN (and FN), PP, and F. Interestingly, the authors also reported a predictive role of the deviation from balanced time perspective (DBTP) [26]. The DBTP has been proposed to measure the individual’s ability to switch from past to future (i.e., to switch between temporal perspectives) in an adaptive way, according to situational changes. Thus, lower values of DBTP indicate a balanced TP, that is, a better ability to switch between all TPs [26]. Finally, Borisenkov et al. [27] showed that poor sleep quality was predicted by higher values of PN and PF, as well as a higher DBTP score. Altogether these studies seem to indicate that the specific nature of an individual’s TP is related to sleep problems [23,24,25,27].

As suggested by Rönnlund et al. [25], in the relationship between TP and sleep disturbance, it is important to consider personality factors. This suggestion could be grounded on, for example, the possible overlap between TPs and personality factors (e.g., neuroticism shows similar characteristics with PN). In addition, a balanced TP is associated with personality traits [28,29,30,31], suggesting a relationship between neuroticism and higher DBTP, and a correlation between high levels of extraversion, openness, agreeableness, and conscientiousness with lower DBTP [28,29,30,31]. Bearing in mind the mixed results when the relation between personality and insomnia has been studied, we set out to further examine the relationship between personality and sleep problems, considering the mediation role of TPs, in a non-clinical sample.

This aim was also based on several discrepant findings reported in the literature and differences between studies concerning, for example, measures and sample characteristics. Indeed, Rönnlund and colleagues used a six-factor version of ZTPI, while Vranesh et al. and Borisenkov et al. used the original five-factor version of ZTPI. Moreover, these studies recruited different populations (from high-school students to university students, as well as adults or older adults), limiting any generalization about the relationship between TP and sleep quality. Related to this point, it is important to take into account the presence of age differences in TP [32] and in sleep quality [33]. Finally, previous studies measured sleep quality (usually using the Pittsburgh Sleep Quality Index or PSQI [34]) instead of insomnia. Although higher scores of PSQI could indicate the presence of sleep problems, the Insomnia Severity Index (ISI) [35,36] seems to be more pertinent to the assessment of insomnia symptoms in individuals [37]. Taking these limitations into account and based on previous studies, our expectation was that conscientiousness would indirectly predict a low ISI score through the mediation role of PP and/or F perspectives. In addition, we expected neuroticism to predict a high ISI score indirectly, through the mediation role of the PN perspective and/or DBTP score. Finally, circadian typology [38] was measured for the purpose of control, given that morningness–eveningness preference has been associated with insomnia [39], personality factors [40], and TP [41].

## 2. Materials and Methods

### 2.1. Participants and Procedure

A sample of 400 volunteers participated in the survey. The participants were unpaid, anonymous, and could withdraw from the study at any time. The participants were recruited through psychological courses (i.e., students responded to an open request for study participation) and advertising posted on social media or using public flyers. The participants filled out the included questionnaires by paper and pencil. Of the participants, 227 were women and 173 were men. The mean age was 36.88 years (SD = 13.29 years; range 18–74 years), and there was no age difference between women (M = 36.65 years; SD = 13.41 years) and men (M = 37.18 years; SD = 13.16 years), with *t*(398) = 0.40, *p* = 0.69, *Cohen’s d* = 0.04. The majority (*n* = 197) of participants reported having a high-school diploma, and 92 individuals had a bachelor’s and/or master’s degree. The remaining participants declared having no school diploma (*n* = 6), elementary school diploma (*n* = 21), or middle school diploma (*n* = 84). Almost 20% of participants said that they did shiftwork, with 26 of these reporting a day–night rotating work schedule, 59 participants reporting a daytime shiftwork schedule, and only 4 participants reporting a fixed nighttime shift. The participants filled in all questionnaires individually after receiving a brief explanation of the study. The study protocol was approved by the Ethical Committee of the Department of Psychology at the University of Campania “Luigi Vanvitelli”, and all participants provided informed consent prior to filling in the questionnaire.

### 2.2. Big Five Inventory—10 Items

Personality was assessed using the short and well-validated 10-item version of the Big Five Inventory (BFI) [42] in its Italian version [43]. The BFI-10 measures each personality trait with two items, one of which is negatively worded and, thus, reverse-scored. The participants rated their personality characteristics on 5-point Likert scales (1 = does not apply at all; 5 = fully applies).The Italian version of BFI-10 identifies 5 personality scales: agreeableness (AG: “*I see myself as someone who…is generally trusting*”); conscientiousness (CON: “*I see myself as someone who…does a thorough job*”); emotional stability (ES: “*I see myself as someone who…get nervous easily*”); extraversion (EX: “*I see myself as someone who…is reserved*”); and openness (OP: “*I see myself as someone who…has an active imagination*”). Guido et al. [42] provided the internal consistency of this 10-item scale in terms of reliability and factor structure, and they proficiently verified its convergent and concurrent validity.

### 2.3. Zimbardo Time Perspective Inventory

To evaluate the TP, we administered the Italian version of the ZTPI [22], adopted by Beracci et al. [41]. This version of the ZTPI contained 56 items rated on a 5-point Likert scale (1 = very uncharacteristic of me; 5 = very characteristic of me). The ZTPI identifies 5 TP dimensions: past-negative (PN: “*I often think about the bad things that have happened to me in the past*”); past-positive (PP: “*I enjoy stories about how things used to be in the good old times*”); present-hedonistic (PH: “*I take risks to put excitement in my life*”); present-fatalistic (PF: “*Since whatever will be will be, it doesn’t really matter what I do*”); and Future (F: “*Meeting tomorrow’s deadline and doing other necessary work comes before tonight’s play*”). The DBTP was considered as an indicator of a balanced TP and was calculated according to the formula reported by Stolarski et al. [23] (see also [41]), reflecting an individual’s ability to switch between TPs (i.e., higher DBTP scores reflect a low level of balanced TPs). The optimal values for each dimension of ZTPI to subtract from observed values are: 1.95 for PN, 3.9 for PH, 4.0 for F, 4.6 for PP, and 1.5 for PF. Beracci et al. [41] provided strong reliability for each scale (Cronbach’s alpha was equal to 0.83, 0.79, 0.85, 0.80, and 0.70, for PN, PP, PH, PF, and F, respectively).

### 2.4. Insomnia Severity Index

The Insomnia Severity Index (ISI) assesses the experience of insomnia symptoms [35,36]. The well-validated Italian version of the ISI comprises 7 items and examines the severity of insomnia symptoms (i.e., satisfaction with current sleep pattern, interference with daily functioning, noticeability of impairment attributed to the sleep problem, and level of distress caused by the sleep problem) over the past two weeks, including difficulty initiating and maintaining sleep, and waking too early. All the items are scored on a 5-point Likert scale (0 = none/very satisfied/not at all interfering/not at all noticeable/not at all; 4 = very/very dissatisfied/very much interfering/very much noticeable/very much), and the total score ranges from 0 to 28, with higher scores representing greater insomnia symptoms. According to the total score and the recommendations provided by Bastien et al. [35], the 0–7 range indicates an absence of insomnia symptoms (i.e., no clinically significant insomnia), 8–14 indicates subthreshold insomnia, the 15–21 range indicates the presence of moderate insomnia, and a range of 22–28 indicates severe insomnia. The Italian version of ISI showed, in a sample of insomnia patients, good psychometric properties due to moderate reliability (Cronbach’s alpha was equal to 0.75 and corrected item-to-total correlations for the items ranged from 0.49 to 0.74), factor structure, concurrent validity, and sensitivity to change in patients after cognitive behavioral treatment for insomnia (CBT-I) [36].

### 2.5. Morningness–Eveningness Questionnaire Reduced Version

As recommended by Rönnlund and Carelli [29] and Rönnlund et al. [30], we administered the reduced version of the Morningness–Eveningness Questionnaire (rMEQ) [44,45] to assess circadian typology. The Italian scale includes 5 questions taken from the original 19-item version of the MEQ [46]. Three items are related to the participants’ preferred time for going to bed, getting up, and the moment of the day when peak personal efficiency is at its maximum. One item requires the participants to assess their degree of tiredness within the first half hour after waking and the final item requires them to indicate which chronotype they think they belong to. The total score ranges from 4 to 25 and higher scores indicate a shift toward morningness. Although Beracci et al. [41] reported a Cronbach’s alpha of below 0.70, this value was in line with the α level reported in the Italian population [47]. More importantly, the concurrent and construct validity of the rMEQ has been demonstrated in an Italian context [45].

### 2.6. Data Analysis

For the statistical analysis, we used the software SPSS version 20 (IBM Corp., Armonk, NY, USA). First, we provided descriptive statistics for each scale, including the mean, standard deviation and range. We also assessed gender differences for circadian typology using a chi-squared (*χ*^2^) test, as well as for age, using a one-way between-subjects ANOVA. Although specific cut-off points for the ISI score are provided [35], for the purpose of this study, an ISI score of 8 or higher indicated insomnia [36]. We then assessed group (non-insomniac vs. insomniac individuals) differences for each variable using a between-subjects *t*-test. Zero-order Pearson correlations between variables were calculated. Finally, to better examine the relationship between personality traits and insomnia, we used the PROCESS macro [48] to undertake a mediation analysis, in which unstandardized indirect effects were obtained from 5000 bootstrap samples; 95% bias-corrected confidence intervals which exclude zero indicate significant indirect/mediation effects. Specifically, in the current study, we used the conceptual model number 4 of Hayes’ templates (see Figure 1) to test whether TPs moderate the effects of conscientiousness or emotional stability (i.e., neuroticism) on the ISI score, controlling for age, gender, and circadian typology. In other words, we estimated the direct/indirect effect of all TPs mediators at the same time (and, thus, we did not perform a multiple mediation analysis for each TP mediator individually).

## 3. Results

All data are reported in the Appendix A. In our sample, we found 14 evening types, 272 neither types, and 114 morning types. No gender differences between each chronotype were found (*χ*^2^(2) = 3.05, *p* = 0.22), while we found that evening types (26.64 ± 5.12 years) were younger than neither types (34.17 ± 11.98 years) and morning types (44.59 ± 13.61 years), as shown by the ANOVA (*F*(2397) = 33.75, *p* < 0.0001, *ƞ*^2^*_p_* = 0.15).

As regards the ISI score, 71.50% of participants reported an ISI score equal to or less than 7, without reporting any insomnia symptoms. The remaining 28.50% of the sample obtained an ISI score that was equal to or higher than 8 (in the present research this percentage represented the “insomnia group”). Specifically, 89 participants reported ISI scores in the subthreshold range, while 25 participants reported ISI scores that reflected moderate or severe insomnia. We did not find gender differences in the distribution of insomnia or normal groups (*χ^2^*(1) = 0.935, *p* = 0.34). In contrast, we found that the insomnia group (39.27 ± 14.14 years) was older than the normal (i.e., good sleepers) group (35.92 ± 12.83 years), with *t*(398) = −2.28, *p* < 0.05, *Cohen’d* = 0.25. At the same time, the insomnia participants (M = 15.72; SD = 3.12) obtained lower rMEQ scores than the normal participants (M = 16.88; SD = 2.80), with a tendency toward eveningness (*t*(398) = 3.61, *p* < 0.0001, *Cohen’s d* = 0.39).

As shown in Table 1, the insomniacs significantly differed from the non-insomniacs for PN, PP, F, and DBTP (with a slight tendency toward significance for PH) when ZTPI was considered, and for CON and EX (with a slight tendency toward significance for ES), when BFI-10 was analyzed. Consequently, the participants with insomnia symptoms were more negatively oriented toward the past, less positively oriented toward the past and future, with a generally less balanced TP. At the same time, they were less conscientious and extroverted.

Table 2 shows the Pearson r correlations between variables. For the purpose of the present study, we observed that ISI score was negatively correlated with PH and F, but it was positively correlated with PN and DBTP. At the same time, the ISI score was negatively correlated with CON, ES, and EX. These results partially mirrored the previous findings in which group comparisons were performed. Additionally, we found negative correlations between PN and ES or EX, while a positive correlation was found between PN and OP. Moreover, we observed positive correlations between CON and F or PP, while PH was positively correlated with EX. Furthermore, PF showed a negative correlation with ES and a positive correlation with EX. Finally, the DBTP was negatively correlated with both CON and ES.

When these correlations were performed controlling for gender, age, and circadian typology, we obtained the same r values in strength and direction with the only exceptions being the following correlations: F and ISI scores (*r* = −0.09, *p* = 0.053); PP and CON (*r* = 0.07, *p* = 0.17); PF and EX (*r* = +0.09, *p* = 0.07); and ISI score and ES (*r* = −0.08, *p* = 0.10).

The mediation model, displayed in Figure 1A, indicated a direct effect of CON on ISI score with *R^2^* = 0.16, *F*(4395) = 18.19, *p* < 0.0001. The *β* value was −0.73 (*t* = −2.69, *p* < 0.05) and 95% CI ranged between −1.26 and −0.20. As shown in Figure 2A, there was an indirect effect: conscientiousness positively predicted the future TP, which in turn negatively predicted the ISI score (*β* = −0.02, 95% CI ranged between −0.04 and −0.002). By contrast, the mediation model of Figure 1B did not report a direct effect of ES on ISI score (*β* = −0.15, *t* = −0.62, *p* = 0.33, 95% CI = −0.61 and +0.32), although the model was significant (*R*^2^ = 0.13, *F*(4395) = 15.16, *p* < 0.0001). However, we found an indirect effect between emotional stability and insomnia mediated by past-negative (*β* = −0.07, 95% CI = −0.12 and −0.03), as shown in Figure 2B. Given that we found differences between insomniacs and non-insomniacs for EX, as well as a negative correlation between ISI and EX, we decided to perform a mediation model using EX as a predictor of ISI score. Not only was a direct effect observed (*β* = −0.76, *t* = −3.32, *p* < 0.05, 95% CI = −1.22 and −0.31) with a significant model (*R*^2^ = 0.18, *F*(4395) = 21.45, *p* < 0.0001), but also an indirect effect of PN (*β* = −0.06, 95% CI = −0.10 and −0.02) and PH (*β* = −0.02, 95% CI = −0.05 and −0.005), as shown in Figure 2C.

## 4. Discussion

The present study examines the relationship between personality traits, according to the BF model, and insomnia symptoms, while assessing the mediation role of time perspectives, and controlling for gender, age, and circadian typology. In a non-clinical sample, we found that insomniacs (defined as all participants with an ISI score ≥ 8), compared to non-insomniacs, reported higher scores for the past-negative aspect, higher scores for the deviation from balanced time perspective, lower future and lower past-positive orientation scores, lower conscientiousness, and lower levels of extraversion. Although a tendency towards significance was found, the insomniacs reported a low PH orientation and low emotional stability. These significant comparisons were confirmed using a continuous approach, as demonstrated by the Pearson correlation analysis. Specifically, the ISI score was positively correlated with both PN and DBTP, while it was negatively correlated with PH and F time perspectives. At the same time, the ISI score was negatively correlated with conscientiousness, emotional stability, and extraversion. Interestingly, we also found that participants with lower past-negative orientations reported higher emotional stability and extraversion levels and lower openness. Participants with lower present-fatalistic orientations reported higher degrees of emotional stability and extraversion. As expected, higher future and past-positive orientations corresponded with higher conscientiousness; a stronger present-hedonistic tendency was associated with higher levels of extraversion. More important, higher DBTP, indicating a diminished ability to shift from past to future, was associated with lower levels of conscientiousness and emotional stability. On one hand, these findings confirm the association between insomnia symptoms and both conscientiousness and emotional stability (i.e., neuroticism) [12,15,16,17,18,19,20,21], with the additional role of extraversion. On the other hand, the results confirm the association between PN, F, and DBTP with poor sleep quality [29,30,31], with the additional role of PH. In addition, we partially confirmed the relationship between conscientiousness and future, and past-positive and balanced time perspective, as well as between neuroticism and past-negative, and present-fatalistic and unbalanced time perspective [24,25,26,27].

As regards personality, these data highlight the predisposing and perpetuating factors of specific personality traits in insomnia, suggesting that emotionally unstable individuals tend to experience worry, rumination, poor coping strategies, and diminished emotion regulation, e.g., [49]. At the same time, all participants with low levels of conscientiousness showed deficits in responsibility, motivation, and self-control [12]. These two personality traits could be in line with the cognitive model proposed by Harvey [50]. According to this model, worry, selective attention monitoring, misperception of sleep, unhelpful beliefs, and safety behaviors could contribute to increased arousal and interfere with sleep. The additional role of extraversion could depend on the mean age of the sample with young adults [51] and on the relationship between extraversion and lower stress reactivity [52], suggesting fewer sleep difficulties [53].

As regards TP, the positive associations between PN and DBTP with the ISI score, on one hand, and the negative associations between PH, PP, and F with the ISI score, on the other hand, could contribute to the major role of thought processes in sleep disorders [50]. Excessive rumination concerns negatively-oriented thinking about the past [54], and it is often recognized as a perpetuating factor in insomnia [50]. In a similar vein, a less balanced TP indicates less ability to switch between temporal dimensions in an adaptive way [26], and DBTP has shown substantial relationships with several measures of well-being [23], including level of perceived stress, which could increase the sleep-related arousal. In line with these assumptions, the fact that lower PH, PP and F were associated with higher frequency in insomnia symptoms could suggest how these TPs are related to health-promoting behaviors and healthy lifestyle habits, such as less frequent use of alcohol and tobacco [22].

Beyond the direct effect displayed in Figure 2, the mediation models could give a more comprehensive picture of the relationship between personality traits and insomnia symptoms. As displayed in Figure 2A, a person with a high level of conscientiousness was likely to be more future-oriented and, in turn, less inclined to develop insomnia symptoms. In line with previous results [22], individuals with a more future-oriented perspective are more optimistic and tend to anticipate positive outcomes. At the same time, the tendency to be reliable, well-organized, and hard-working (i.e., conscientiousness) was associated with good sleep quality and fewer insomnia symptoms [9,10,11,12,13]. Thus, the future time perspective, related to a higher degree of conscientiousness, induces the perpetuation of more positive functioning and more adaptive coping strategies [22], reducing the probability of reporting sleep problems. In a mirroring manner, Figure 2B shows that a person with a lower level of emotional stability was more past-negative-oriented, and, in turn, reported insomnia symptoms. In other words, a high level of neuroticism can be associated with anxiety, depression, and stress-perceived levels, which may induce insomnia [55]. At the same time, the PN time perspective, associated with a negative, antipathetic, and ruminative orientation toward the past, was related to increased levels of stress, arousal, and tension [56], which could induce insomnia [50]. Finally, Figure 2C indicates two indirect paths: (1) high extraversion was associated with less of a PN orientation which, in turn, was associated with a higher ISI score; (2) high extraversion was associated with a strong PH orientation, which, in turn, was related to a lower ISI score. The first path could suggest that extraversion, a trait defined by characteristics such as being outgoing, sociable, and experiencing positive emotions [57], reduced the rumination related to negative past events and the pessimistic or aversive attitude toward the past. This reduction could decrease the tendency to experience more negative affect, increasing stress reactivity [52,53] and promoting a more physically active lifestyle [51,58,59]. Thus, in turn, sleep quality was better, or, in other words, participants experienced fewer insomnia symptoms [59]. Although less intuitive, the second path could reflect the fact that extraversion induces a more positive affect, which is typically experienced by individuals oriented toward PH. Although the PH scale describes a hedonistic, risk-taking attitude toward time and life, with pleasure and enjoyment in the present without any consideration for future outcomes, it is important to bear in mind that Stolarski et al. [23] reported an ideal moderately high score (i.e., 3.9) on the PH dimension for the calculation of DBTP. As reported by Wiberg et al. [60], the PH scale describes participants who are very active, have a wide range of interests, and take part in a variety of events and activities. PH-oriented participants have a wide social network, are very socially engaged, have deep friendships and take part in group activities, are interested in being outside and enjoying nature and have time for themselves, staying at home and reflecting in solitude. This profile probably induced positive affect, a high mood state, and better management of stress. Thus, it was not surprising that a high PH score was associated with a lower ISI score. The experience of more positive affect was related to less stress reactivity and a better quality of sleep.

However, the present study was not immune to some limitations. Although we performed comprehensive measurements of the variables in a relatively large population-based sample, the data were cross-sectional, and no causal links could be advanced. Longitudinal studies are needed to investigate the causal links between personality, time perspective, and insomnia. In addition, we pointed out that the link between personality and insomnia is bi-directional. In other words, insomnia severity (e.g., subjective experience of non-restful sleep, feeling tired during the day, etc.) could increase the perceived stress, reducing the ability to cope with stress and promoting neuroticism. At the same time, a positive subjective sleep experience and healthy sleep habits could induce less stress and better coping abilities, as well as develop self-control, promoting conscientiousness. In addition, a specific model [61] highlights the importance of both physiological markers to insomnia and psychological precursors, leading, respectively, to physiological and cognitive arousal, which is linked to the degree of being able to respond to or cope with stressful situations, probably causing certain personality traits [15,62]. Although ISI was more appropriate to detect insomnia in the population than PSQI [34,35,36], ISI is not a diagnostic tool, but merely a means of screening. Thus, these data should be confirmed in a clinical insomnia population with the additional objective recording of sleep and sleep behavior. For instance, the polysomnography (PSG) has been recommended by the European Sleep Research Society [63] to capture in a more objective and accurate way the increased arousal level, especially during REM (rapid eye movements) stage, according to the hyperarousal model of insomnia [50]. At the same time, the PSG can provide specific sleep measures for sleep quality to be used as markers of the subjective sleep quality experience [64]. Finally, we did not control for other variables, such as anxiety and depression, although our data were controlled for important variables, all related to personality, time perspective, and insomnia. Future studies should consider the mood levels of participants. Although we took note of the daily and/or night-shift work schedule of our participants, we did not take note of their supplements and/or medications to manage their insomnia. This information could be considered as a potential confound in future studies. Finally, we acknowledge that in our sample the number of evening-types was small, probably due to the mean age of the participants.

## 5. Conclusions

In a cross-sectional study, the present research demonstrates, in a non-clinical sample, that specific personality traits and time perspectives are related to insomnia symptoms. Specifically, conscientiousness and extraversion were lower in people who obtained at least 8 in the ISI, and this was associated with lower past-positive and future time perspectives. At the same time, higher past-negative and DBTP were associated with insomnia symptoms. Moreover, the mediation analyses indicated that conscientiousness indirectly predicted the ISI score through the future orientations. In addition, the analyses showed that emotional stability (i.e., neuroticism) indirectly predicted insomnia symptoms through the PN perspectives. Finally, extraversion was associated with ISI score through the mediation role either of PN or of PH time perspectives. These data could have practical implications regarding clinical interventions. Specific personality traits could be highlighted as a risk factor for dropout from and treatment resistance to cognitive behavioral therapy for Insomnia (CBT-I) [65]. At the same time, time perspective therapy (TPT) [66] could be an additional treatment for sleep-related problems, given that the therapist’s goal is to establish a balanced TP of past, present, and future. Finally, it has recently been proposed that mindfulness-based interventions [67], or mindfulness traits [68], are effective in reducing sleep problems. Future studies should confirm these data in longitudinal studies in clinical populations, and they should verify the benefits of specific clinical treatments. Future studies should also address the possibility that our mediation models can fit in a different way with specific insomnia types.

## Figures and Tables

**Figure 1 ijerph-19-11018-f001:**
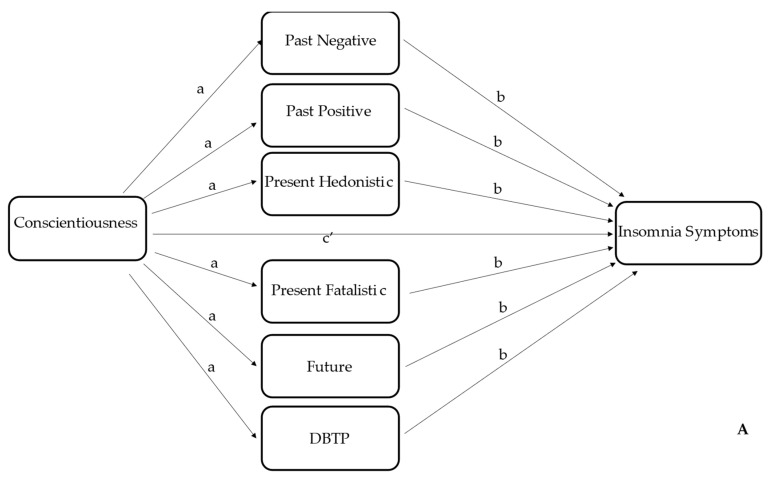
The mediation model tested. The a and b parameters indicate an indirect effect, while the c parameter indicates a direct effect. (**A**) the model tested the relationship between consciousness and insomnia symptoms through the mediation effect of each TPs and DBTP. (**B**) the model tested the relationship between emotional stability and insomnia symptoms through the mediation effect of each TPs and DBTP. Note that in both models the mediation analysis considered all TPs mediators at the same time.

**Figure 2 ijerph-19-11018-f002:**
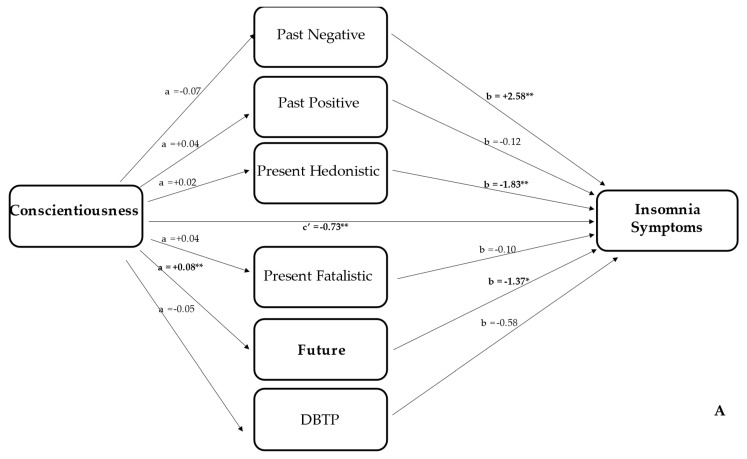
(**A**) The mediation model with conscientiousness as a predictor of ISI score with all TPs as mediators; (**B**) the mediation model with emotional stability as a predictor of ISI score with all TPs as mediators; (**C**) the mediation model with extraversion as a predictor of ISI score with all TPs as mediators. In all figures, the significant relationships are in bold; * indicates *p* < 0.05, while ** indicates *p* < 0.0001.

**Table 1 ijerph-19-11018-t001:** The mean (and its SD) for every variable, separately for insomnia and normal groups. The values of *t* and *p* of all between-subjects *t*-test comparisons are also provided. In bold, the significant comparisons.

ZTPI	Insomnia Group (*n* = 144)	Normal Group (*n* = 286)	Between-Subjects *t*-Test
**PN**	**3.09 (0.68)**	**2.82 (0.62)**	***t*(398) = −3.82, *p* < 0.0001, *Cohen’s d* = 0.41**
**PP**	**3.52 (0.52)**	**3.66 (0.53)**	***t*(398) = +2.38, *p* < 0.05, *Cohen’s d* = 0.27**
PH	3.14 (0.59)	3.26 (0.58)	*t*(398) = +1.85, *p* = 0.065, *Cohen’s d* = 0.21
PF	2.81 (0.60)	2.80 (0.59)	t(398) = −0.21, *p* = 0.83, *Cohen’s d* = 0.02
**F**	**3.31 (0.43)**	**3.50 (0.45)**	***t*(398) = +3.38, *p* < 0.005, *Cohen’s d* = 0.43**
**DBTP**	**2.54 (0.62)**	**2.29 (0.55)**	***t*(398) = −3.95, *p* < 0.0001, *Cohen’s d* = 0.43**
**BFI-10**	**Insomnia Group (*n* = 144)**	**Normal Group (*n* = 286)**	**Between-Subjects *t*-Test**
Agreeableness	3.16 (0.88)	3.33 (0.85)	*t*(398) = +1.77, *p* = 0.08, *Cohen’s d* = 0.20
**Conscientiousness**	**3.55 (0.91)**	**4.00 (0.79)**	***t*(398) = +5.00, *p* < 0.0001, *Cohen’s d* = 0.53**
Emotional Stability	2.81 (0.99)	3.01 (0.95)	*t*(398) = +1.87, *p* = 0.06, *Cohen’s d* = 0.21
**Extraversion**	**3.14 (0.92)**	**3.59 (0.99)**	***t*(398) = +4.10, *p* < 0.0001, *Cohen’s d* = 0.47**
Openness	3.18 (0.98)	3.16 (0.92)	*t*(398) = −0.12, *p* = 0.90, *Cohen’s d* = 0.02

Abbreviations: PN for past-negative; PP for past-positive; PH for present-hedonistic; PF for present-fatalistic; F for future; DBTP for deviation from balanced time perspective.

**Table 2 ijerph-19-11018-t002:** The Pearson r correlation of zero-order correlations between variables. In bold, the significant correlations.

	PN	PP	PH	PF	F	DBTP	ISI	AG	CON	ES	EX	OP
**PN**	**1**	−0.07	**+0.21 ****	**+0.43 ****	**+0.11 ***	**+0.59 ****	**+0.26 ****	−0.03	−0.09	**−0.20 ****	**−0.19 ****	**+0.16 ***
**PP**	**-**	**1**	−0.06	−0.03	**+0.29 ****	**−0.51 ****	**−0.08**	+0.06	**+0.11***	−0.01	−0.03	−0.02
**PH**	**-**	**-**	**1**	**+0.47 ****	+0.003	+0.03	**−0.15 ***	−0.03	−0.003	−0.08	**+0.18 ****	+0.08
**PF**	**-**	**-**	**-**	**1**	−0.10	**+0.59 ****	+0.07	−0.07	+0.003	**−0.17 ***	**+0.10 ***	−0.01
**F**	**-**	**-**	**-**	**-**	**1**	**−0.30 ****	**−0.15 ***	−0.01	**+0.20 ***	−0.02	+0.01	+0.03
**DBTP**	**-**	**-**	**-**	**-**	**-**	**1**	**+0.23 ****	−0.07	**−0.12 ***	−0.11 *	−0.09	+0.04
**ISI**	**-**	**-**	**-**	**-**	**-**	**-**	**1**	−0.03	**−0.25 ****	**−0.11 ***	**−0.26 ****	−0.02
**AG**	**-**	**-**	**-**	**-**	**-**	**-**	**-**	**1**	−0.01	**+0.26 ****	**−0.13 ***	**−0.21 ****
**CON**	**-**	**-**	**-**	**-**	**-**	**-**	**-**	**-**	**1**	+0.08	**+0.23 ****	+0.01
**ES**	**-**	**-**	**-**	**-**	**-**	**-**	**-**	**-**	**-**	**1**	**−0.15 ***	**−0.25 ****
**EX**	**-**	**-**	**-**	**-**	**-**	**-**	**-**	**-**	**-**	-	**1**	+0.05
**OP**	**-**	**-**	**-**	**-**	**-**	**-**	**-**	**-**	**-**	-	**-**	**1**

Note * *p* < 0.05 and ** *p* < 0.0001. Abbreviations: PN for past-negative; PP for past-positive; PH for present-hedonistic; PF for present-fatalistic; F for future; DBTP for deviation from balanced time perspective; AG for agreeableness; CON for conscientiousness; ES for emotional stability; EX for extraversion; OP for openness.

## Data Availability

All data are in the Appendix A.

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
