# Peer review of "Insomnia, Time Perspective, and Personality Traits: A Cross-Sectional Study in a Non-Clinical Population"

_ijerph, 2022, doi:10.3390/ijerph191711018_

Round 1
Reviewer 1 Report
Insomnia can be considered a major concern that disrupts the quality of life, work performance, and public health in general. In the submitted manuscript, the authors examine personality traits and chronotypes that may influence insomnia. Although the topic of insomnia is related more to the field of healthcare, the content can be of interest to general public health and is suitable for the journal. The strength of this paper includes a clear objective, good sample size and mediation analysis. Future clinical practice must consider the finding of this study as the problem of insomnia is rampant.
Some feedback for improvements can be made.
1) Mediation Analysis: the authors mentioned model 4 of Hayes’. But since the model presented here considers all mediating variables together at once, can the authors clarify in the Figure caption or Methods that the direct/indirect effects are estimated considering all mediators “at once”. If not, one wonders why not just perform multiple model 4 for each mediator of interest individually.
2) Minor: Statistical software for stats test shall be mentioned in the Methods (SPSS?).
3) Participants: Do the authors consider night-shift as a confound in their analysis? This is because insomnia can be a problem for those people, which has nothing to do with traits. Do the authors also take note of their supplements and/or medication to manage their insomnia?
4) The authors may consider discussing polysomnography as a more objective/accurate way for assessing insomnia instead of ISI, which is categorical in nature, and thus not as sensitive (this is usually a case for many clinical ax. tools). Other studies have used polysomnography to measure sleep quality.
Author Response
Reviewer#1:
“English language and style: English language and style are fine/minor spell check required”.
This manuscript was revised by a mother English tongue in its original version. Considering the reviewer#1’s suggestion, a second round of English revision was performed.
“Comments and Suggestions for Authors:
Insomnia can be considered a major concern that disrupts the quality of life, work performance, and public health in general. In the submitted manuscript, the authors examine personality traits and chronotypes that may influence insomnia. Although the topic of insomnia is related more to the field of healthcare, the content can be of interest to general public health and is suitable for the journal. The strength of this paper includes a clear objective, good sample size and mediation analysis. Future clinical practice must consider the finding of this study as the problem of insomnia is rampant. Some feedback for improvements can be made”.
We would like to thank the reviewer for his/her positive comments of the manuscript. Also, we really appreciated for selecting “yes” option for all points presented by the Journal review form. We took into account all points raised by the reviewer#1 and all changes in the text were made in red.
“1) Mediation Analysis: the authors mentioned model 4 of Hayes’. But since the model presented here considers all mediating variables together at once, can the authors clarify in the Figure caption or Methods that the direct/indirect effects are estimated considering all mediators “at once”. If not, one wonders why not just perform multiple model 4 for each mediator of interest individually”.
The reviewer#1 is right. We did not perform multiple model 4 for each mediator of interest individually, but, instead, we performed the model 4 at once with all mediators at once. In the method -section (page 5, lines 220-223) and in the Figure 1 caption (page 6, lines 227-228), we mentioned this aspect.
“2) Minor: Statistical software for stats test shall be mentioned in the Methods (SPSS?)”.
We agreed with the reviewer#1 and we reported the statistical software used for the stats in the method section (page 4, line 206).
“3) Participants: Do the authors consider night-shift as a confound in their analysis? This is because insomnia can be a problem for those people, which has nothing to do with traits. Do the authors also take note of their supplements and/or medication to manage their insomnia?”
The reviewer#1 raised two important aspects related to insomnia, especially as potential confound in our analysis. As regards the night-shift, at page 3, in the description of participants, we explicitly indicated that only 4 participants (out of 400) reported a night-shift work. Thus, we did not think that this variable was a confound. As regards the second aspect, we did not take note of supplements and/or medication to manage insomnia. We decided to highlight this second aspect as a limit of our research in the discussion (page 12, lines 423-426).
“4) The authors may consider discussing polysomnography as a more objective/accurate way for assessing insomnia instead of ISI, which is categorical in nature, and thus not as sensitive (this is usually a case for many clinical ax. tools). Other studies have used polysomnography to measure sleep quality”.
We agreed with the reviewer#1 that the polysomnography (PSG) is the gold standard, not only to study sleep quantitative, but also sleep quality. In the discussion, at page 12, lines 394-397, we highlighted the limit of the ISI (we only advanced the idea that ISI can be better than PSQI in assessing/screening insomnia) and we specified that objective measure is needed to assess clinical insomnia. However, we added a sentence, indicating that PSG is a powerful tool for assessing insomnia (page 12, lines 415-420).
Reviewer 2 Report
This is a well-written paper, regarding a not novel but always in the list of topics about insomnia. Personality traits are interesting but challenging to investigate because of the heterogenicity of the assessment tools.
In my opinion, introduction is too long and difficult to follow at times. I would consider summarizing it and review some of the abbreviations included, as they make the lecture not fluent. Introduction should be more focus on how the authors will justify their investigation and results. I miss this perspective.
A list of abbreviations at the very beginning could be useful.
Please, clarify how were the participants recruited.
Consider including a figure summarizing the information contained in table 1, in order to make it more visual.
Author Response
Reviewer#2
“English language and style: English language and style are fine/minor spell check required”.
This manuscript was revised by a mother English tongue in its original version. Considering the reviewer#2’s suggestion, a second round of English revision was performed.
“This is a well-written paper, regarding a not novel but always in the list of topics about insomnia. Personality traits are interesting but challenging to investigate because of the heterogenicity of the assessment tools”.
We thank the reviewer#2 for her/his positive comment about the manuscript. We acknowledged that the topic is not new, but we think that the manuscript can contribute to the topic introducing a new mediating variable (i.e., Time Perspective), with potential clinical practices. In addition, we referred to the Big Five model, adding further convergent results in the relationship between personality and insomnia, although different tools have been used to assess personality. We took into consideration all reviewer#2’s suggestions and all changes in the text were in red.
“In my opinion, introduction is too long and difficult to follow at times. I would consider summarizing it and review some of the abbreviations included, as they make the lecture not fluent. Introduction should be more focus on how the authors will justify their investigation and results. I miss this perspective”.
We thank the reviewer#2 for this suggestion. We modified the introduction, trying to shorten it and to focus on justification and perspective of the study. The abbreviations were reviewed when it was possible.
“A list of abbreviations at the very beginning could be useful”.
We inserted a list of abbreviations before the introduction, as requested. We decided to leave the abbreviations for all five personality factors in the method section and for the Table2, while we decided to explicit all personality factor in the Table1 in order to make this Table more informative. This change should be accepted by the Editorial staff of the Journal.
“Please, clarify how were the participants recruited”.
We thank the reviewer#2 for this point, and we added a sentence in the participant’s description about the recruitment (page 3, lines 127-130).
“Consider including a figure summarizing the information contained in table 1, in order to make it more visual”.
Probably the reviewer#2 suggested an important point, making the results in the Table 1 more visual. However, we think that the Table 1 summarized the results of group comparisons for ZTPI factors and BFI factors. All significant results in the Table 1 were in bold, easily making their identification. In addition, we think that the Figures in the text were adequate to display the results (a little bit more complicated than group comparison) of the mediation analysis. Thus, we decided to leave the Table1.
Reviewer 3 Report
This is a well-designed and conducted study. The study shows an association between personality traits and insomnia. The authors are attributing these personality traits as being a causal association for insomnia. Although this may be the case, it is also possible that insomnia may contribute to causing some of these personality traits. Insomnia may instead have a physiological basis, thereby causing certain personality traits. It would be useful to recognize and discuss this possibility in the Discussion.
Author Response
Reviewer#3
“English language and style: English language and style are fine/minor spell check required”.
This manuscript was revised by a mother English tongue in its original version. Considering the reviewer#3’s suggestion, a second round of English revision was performed.
“This is a well-designed and conducted study. The study shows an association between personality traits and insomnia. The authors are attributing these personality traits as being a causal association for insomnia. Although this may be the case, it is also possible that insomnia may contribute to causing some of these personality traits. Insomnia may instead have a physiological basis, thereby causing certain personality traits. It would be useful to recognize and discuss this possibility in the Discussion”.
We thank the reviewer#3 for her/his positive comments about the paper. In addition, we thank the reviewer#3 for choosing “yes” option for almost all points presented in the Journal review form. We agreed with the reviewer#3 that the relationship between personality traits and insomnia is bi-directional. Although, in the manuscript, we stressed a specific directional model (personality à TP à insomnia), in the original version of the paper we acknowledged, as limit, the nature of the study (cross-sectional), limiting any casual inference (page 12, lines 401-403). However, we recognized the reviewer#3’s suggestion in the discussion (page 12, lines, 403-412). All changes, in the text, were made in red.